# The Detection Algorithm Based on the Short Time Delay for Three-Phase Unbalanced Voltage Sag

**Xiaohong Hao \*, Yangtao Wang, Shuaizheng Sun, Shan Jiang and Yan Liu**

School of Mechanical and Electronical Engineering, University of Electronic Science and Technology of China, Chendu 611731, China
\* Correspondence: haoxiaohong@uestc.edu.cn; Tel.: +86-028-6183-0229

**Abstract:** To improve the detection speed of the three-phase unbalanced voltage sag, the traditional $\alpha\beta/dq$ transformation algorithm can be applied. However, in this algorithm, the virtual quadrature voltage signal (VQVS) needs to be accurately constructed in the $\alpha\beta$ coordination system. Then, the information of the voltage jump can be achieved by the $\alpha\beta/dq$ transformation. However, the traditional algorithm for constructing the VQVS introduces the harmonic component, so that the detection speed and accuracy is low. To solve the problem, this article modifies the algorithm and presents a new algorithm based on the combination of the short time delay (STD) and the positive- and negative-sequence transformation (PNST) algorithm. The STD algorithm and its construction are firstly detailed. Then, the extraction step and performance of the PNST are given. Since the STD algorithm can avoid harmonic properties, voltage sag signal can be quickly obtained by analyzing the positive and negative sequence component derived from the PNST algorithm. The proposed algorithm is compared with two alternatives already proposed in the literature in terms of detection time. Finally, simulation and experimental results validate the validity of the proposed algorithm.

**Keywords:** three-phase unbalanced voltage; short time signal delay (STD); positive and negative sequence transformation (PNST)

## 1. Introduction

Voltage sags caused by faults in power distribution networks are the most common power quality problem [1]. However, it has obvious influences on the security and stability of the electrical equipment with sensitive loads, such as computers, micro-controllers and new power electronic equipment [2]. As a result, it is becoming an important research topic. The typical hardware of the Dynamic Voltage Restorers (DVRs) provides a cost-effective solution to protect sensitive loads not only against voltage sag but also against those additional disturbances [3–5]. Its compensation strategy requires a fast and robust sag detection algorithm.

Numerous algorithms focusing on the voltage sag detection have been presented in the available scientific literatures. In time domain, the typical voltage sag detection algorithms include Root mean square (RMS) value of the grid voltage, fundamental component algorithm, single-phase voltage transformation average algorithm, peak value voltage, defective voltage algorithm, and so on [6]. However, the above algorithms have some limitations, such that the RMS algorithm and the fundamental component algorithm can only detect the amplitude, not the phase. What's more, their speed is also slow. The single-phase voltage transformation average algorithm is sensitive to noise and high harmonics, and the detection results are not accurate although the filter is used in the system. The peak voltage algorithm can accurately measure the start and end time of the voltage sag, but when the amplitude of the voltage sag and the rated voltage is small, the detection results have a great deviation from the actual value. The deficient voltage algorithm can be used to detect voltage sag by the estimated value and measured value, which can't detect the definite time of start and end of the voltage sag.

In order to overcome the limitations of the above algorithms, the instantaneous reactive power theory is applied to detect the voltage sag. The instantaneous voltage *dq* transformation algorithm is proposed in [7]. Constructing the virtual voltage signals by delaying T/6 sampling time and getting the voltage amplitude and phase. However, it needs a long detection time. In Ref. [8], the above algorithm is improved by using the difference algorithm on constructing the virtual three-phase voltage signals. The algorithm improves the detection speed, but it makes the noise and high frequency harmonic component amplified. In Ref. [9], a new detection algorithm is discussed, which is proposed to construct the VQVS by delaying T/4 sampling time and get the amplitude and phase using the coordinate transformation from the two-phase stationary *αβ* coordinate system to the rotating *dq* coordinate system. It alleviates this computational burden of the above two algorithms, but it introduces the T/4 delay time. The algorithm to construct the VQVS by using the difference algorithm based on the *αβ/dq* transformation is discussed in [10]. It can detect the voltage sag without delay, but it still can make the noise and high frequency harmonic component amplified [5]. In Refs. [11,12], the algorithm which is proposed to construct the VQVS by delaying a shorter sampling time that less than T/4. Its purpose is to improve the detection speed and anti-interference ability simultaneously. But the result is not ideal. The above detection algorithms, such as delay T/4, delay T/6, difference algorithm, and so on, are very effective for the single-phase voltage sag or three-phase symmetrical voltage sag detection, but they are not suitable for the three-phase unbalanced voltage sag detection. Also, the detection speed can't meet the requirement of the DVRs. A voltage sag detector for a DVR that provided fast transient responses (3–4 ms approximately) is presented in [13]. In Ref. [14], another algorithm is presented to classify power quality disturbances based on symmetrical component decomposition, showing its potential to reject disturbances. However, it is slow compared to other algorithms. A positive-and negative-sequence current decomposition algorithm that includes a quadrature signal estimation algorithm is presented in [15]. A similar strategy was proposed in [16], which is to generate orthogonal signals, and it has successful results. In Ref. [17], the quadrature signal algorithm and the module function are used to provide Delayed Signal Cancellation properties, which make the sag detector adapt to the frequency variations. It is shown that this algorithm provides harmonic cancellation properties when the parameters are adequately tuned. The detection algorithm of the positive sequence and negative sequence based on the double *dq* transformation is proposed in [18]. its advantage is no delay. a novel algorithm based on the combination of least error squares filters and an improved instantaneous symmetrical components algorithm are proposed to improve the dynamic response of the DVR in [13].

The traditional algorithms for detecting voltage sag in the frequency domain include the classical Fourier transform [19,20], Short-Time Fourier transform (STFT), S-transform, Hilbert-Huang transform, and so on. The short-time Fourier transform (STFT) is an improvement of the fast Fourier transform (FFT) technique [21]; however, the performance of the STFT algorithm depends on the choice of the window and there is a compromise between frequency and time resolution [22]. Therefore, the possibility of using a new signal processing tool, based on different principles has been explored over the past decade, with the purpose of overcoming the abovementioned limitations. In Refs. [23,24], the Short-Time Fourier Transform (STFT) detection algorithm is proposed, which gets the voltage amplitude and phase by the localization of the short window function. The advantage is that it can perform local analysis of the signal that compensated the shortcomings of the conventional Fourier transform, and the disadvantage is that it can only detect the voltage within the fixed scale, because it performs signal selection by the shape and the window width of the short window function. In order to overcome the limitations of the STFT detection algorithm, the wavelet transform theory which can process weak signals as well as unstable signals effectively is proposed in [25–27]. It has the advantage that the telescopic and translation operations for the different frequencies component can be performed, so that the details of the signals can be dealt with and the detection accuracy

may be improved. In Ref. [28], the wavelet transform theory is combined with the effective value detection algorithm of the traditional time-domain transform to improve the accuracy of the effective value detection algorithm based on the maxima principle of the wavelet transform modulus.

Although the wavelet transform algorithm has cost is high because of its huge calculation amount, and the harmonic and noise suppressing also are not enough [29,30]. Later, the S-transform theory is proposed by combining the STFT theory with the wavelet transform theory [31]. The S-transform has the advantages both wavelet transform and STFT. However, because its gaussian window function can't be used to adjust the time-frequency resolution, so an improved S-transform detection algorithm with certain noise immunity is proposed in [30], which can quickly detect the characteristic information of the voltage sag. The S-transform detection algorithm is improved, and a algorithm based on discrete orthogonal S-transform is proposed in [29]. Here, the fundamental frequency time-amplitude curve and frequency-amplitude curve are discreted to obtain amplitude and phase information of the voltage sag. It not only can detect voltage sag with compound disturbances quickly and accurately, but also can effectively suppress the noise in the signal. A modified S transform with digital prolate spheroidal window (DPSW) is proposed in [32]. The DPSW has superior time-frequency distribution, so that it can get some characteristics of voltage sag signal, such as sag depth, duration, and phase jump. Compared with the traditional voltage sag detection algorithms, the proposed algorithm has the advantages of high accuracy, low computational complexity, and strong anti-interference ability. Nevertheless, it's a pity that there is no detection time. In Ref. [33], it has proposed the application of the Hilbert–Huang transform (HHT) algorithm for decomposition of the PQ data into its individual frequency components in order to separate the nonstationary voltage sag waveform containing the fundamental frequency component. After the decomposition of the data using the HHT algorithm, an innovative fundamental frequency-based algorithm is proposed for the accurate detection of voltage sag starting and ending times. The proposed algorithm is further extended for automatic detection of the transition segments to accurately calculate the time-varying and single event characteristics such as voltage sag amplitude and phase-angle jump.

According to the opened literatures, there are two kinds of quadrature component construction algorithms, they are separately the delayed T/4 algorithm and the difference algorithm. The two algorithms have the disadvantage of the high delay and low anti-noise ability. Therefore, in this article which not only can provide the quadrature signal quickly and accurately, but also can reject the noise disturbance.

To solve the above problem, a new quadrature component construction algorithm based the combination of the STD and PNST is introduced; this approach is realized by designing a new algorithm constructing the virtual quadrature component and extracting the positive sequence and negative sequence component. Thus, the detection speed problem can be improved. The main contributions of this article are summarized as follows.

(1) The detection principle and prediction model for the three-phase unbalance voltage sag are established.
(2) The new algorithm for constructing the VQVS and many advantages, it also has some disadvantage such that the the steps for extracting the positive sequence and negative sequence component are explained.
(3) To highlight the performance of the proposed algorithm, it has been compared with two other already proposed in the literature. It is shown that other sag detectors can provide robustness against voltage disturbances. However, the one proposed here can do it with a straightforward design and a fast transient response.

The rest of this article is organized as follows. Section 2 presents the detection principle of the three-phase unbalance voltage sag. In Section 3, the STD for constructing the VQVS is proposed. To improve the detection speed and accuracy, the VQVS is reconstructed by assuming the sampling cycle as the delay time. Step-by-step demonstrations on the accuracy of the VQVS are shown. Furthermore, both difference algorithm and STD algorithm are

compared. In Section 4, the extracting step of the positive sequence and negative sequence components are shown. Section 5 shows the simulation and experimental results to validate the proposed algorithm. Finally, the conclusion is summarized in Section 6.

## 2. The Principle of the Detection Algorithm

The algorithm proposed in this article is depicted in Figure 1; the system mainly consists of the STD, PNST, and $\alpha\beta/d_q$ transformation the three-phase grid voltage signal at a certain time $U_{abc}(k)$ is collected and low-pass filtered (Pre-LPF); each phase voltage signal in the three-phase voltage is separated, and the VQVS corresponding to each phase voltage are constructed by the STD separately. Then, the PNST are performed for the original voltage signal and the VQVS to obtain the positive sequence component $U^+{}_{abc}(k)$ and negative sequence component $U^-{}_{abc}(k)$. Lastly, the synchronous coordinate transformation is performed to obtain the amplitude and phase of the voltage signal. The STD and the PNST will be explained as follows.

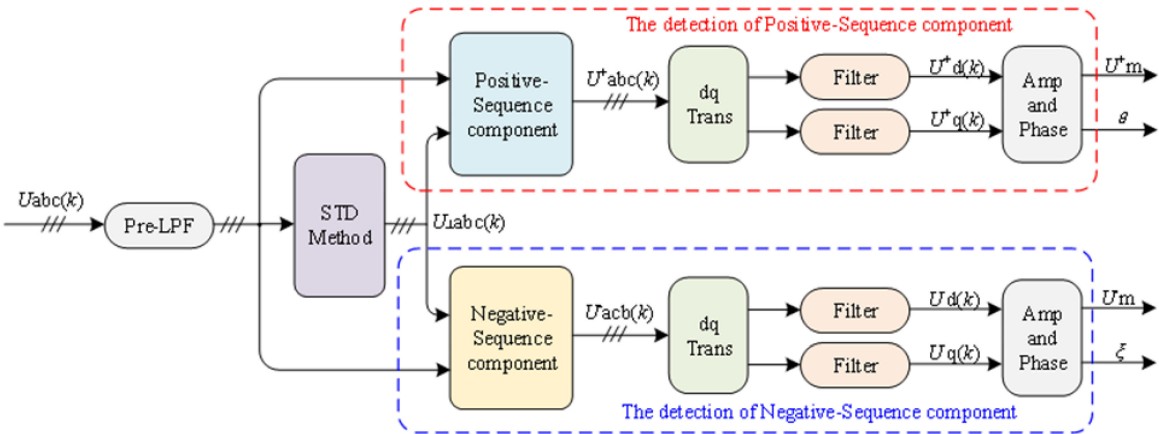

**Figure 1.** The schematic diagram of the three phase unbalanced voltage sag drop detection algorithm.

## 3. Short Time Delay (STD) Algorithm

### 3.1. Basic Principle

The typical step of detecting the amplitude and phase of the voltage vector is to transform the projections $u_\alpha$ and $u_\beta$ of the voltage vector $U_m$ under the stationary coordinate system $\alpha\beta$ to the synchronous rotating coordinate system $d_q$, then the projections are changed as $u_d$ and $u_q$. The process of the coordinate transformation is as follows.

The coordinate system including static $\alpha\beta$ axis and rotating $dq$ axis is shown in Figure 2, the d-axis of the $dq$ coordinate system coincides with the $\alpha$-axis of the $\alpha\beta$ coordinate system at the initial moment, the initial phase angle between the voltage vector $U_m$ and the d-axis in the $dq$ coordinate system is $\varphi$, and $U_m$ and the $dq$ coordinate system rotate counterclockwise with the angular velocity $\omega$, simultaneously.

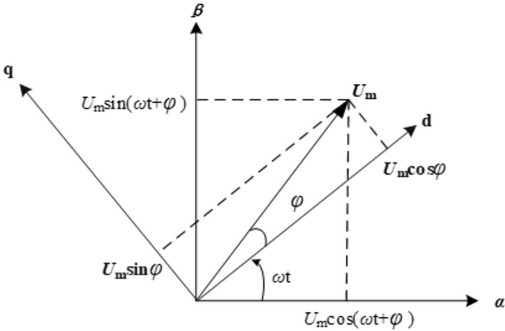

**Figure 2.** The schematic diagram of $\alpha\beta/dq$ coordinate transformation.

The projections of the voltage vector in the $dq$ coordinate system can be given by

$$\begin{cases} u_d = U_{\mathrm{m}} \cos \varphi \\ u_q = U_{\mathrm{m}} \sin \varphi \end{cases} \tag{1}$$

The projections of the voltage vector in the $\alpha\beta$ orthogonal coordinate system can be given by

$$\begin{cases} u_\alpha = U_{\mathrm{m}} \cos(\omega t + \varphi) \\ u_\beta = U_{\mathrm{m}} \sin(\omega t + \varphi) \end{cases} \tag{2}$$

where, $u_\alpha$, $u_\beta$ represent the voltage component of the single-phase voltage on the $\alpha$-axis and $\beta$-axis respectively.

According to the Equations (1) and (2), the transformation from $\alpha\beta$ orthogonal coordinate system to $dq$ coordinate system can be derived as.

$$\begin{cases} u_\alpha = U_{\mathrm{m}} \cos(\omega t + \varphi) \\ u_\beta = U_{\mathrm{m}} \sin(\omega t + \varphi) \end{cases} \tag{3}$$

Then the dc components $U_{d0}$ and $U_{q0}$ in the $dq$ coordinate system are obtained through the low-pass filter, so the amplitude and phase jumps of the fundamental voltage can be obtained as.

$$U = \sqrt{U_{d0}{}^2 + U_{q0}{}^2} \tag{4}$$

$$\varphi = \arctan \frac{U_{q0}}{U_{d0}} \tag{5}$$

If the sampled voltage signal is set as $u_\beta$, then the VQVS needed to be constructed is $u_\alpha$.

Assuming the delay time is one sampling period $T_{\mathrm{s}}$, then the voltage vector $U$ in the $\alpha\beta$ coordinate system can be written.

$$\begin{cases} u_\alpha(t) = U \cos[\omega(t + T_{\mathrm{s}}) + \varphi - \omega T_{\mathrm{s}}] \\ u_\beta(t) = U \sin[\omega(t + T_{\mathrm{s}}) + \varphi - \omega T_{\mathrm{s}}] \end{cases} \tag{6}$$

where, $U$ is the amplitude of the voltage vector; $\varphi$ is the initial phase of the voltage vector; $u_\alpha(t)$ and $u_\beta(t)$ are respectively the projection of the voltage vector on the coordinate axis. Equation (6) can be further written.

$$\begin{cases} u_\alpha(t) = u_\alpha(t + T_{\mathrm{s}}) \cos(\omega T_{\mathrm{s}}) + u_\beta(t + T_{\mathrm{s}}) \sin(\omega T_{\mathrm{s}}) \\ u_\beta(t) = u_\beta(t + T_{\mathrm{s}}) \cos(\omega T_{\mathrm{s}}) - u_\alpha(t + T_{\mathrm{s}}) \sin(\omega T_{\mathrm{s}}) \end{cases} \tag{7}$$

Simplify Equation (7), the relationship between $u_\alpha(t)$ and $u_\beta(t)$ under delaying a sampling period $T_{\mathrm{s}}$ can be written as.

$$u_\alpha(t) = \frac{u_\beta(t + T_{\mathrm{s}}) - u_\beta(t) \cos(\omega T_{\mathrm{s}})}{\sin(\omega T_{\mathrm{s}})} \tag{8}$$

Assuming that one phase of the grid voltage signal detected is $u_\beta(t)$, then the $u_\alpha(t)$ constructed by the VQVS can be expressed as.

$$\begin{aligned} u_\alpha(t) &= \frac{u_\beta(t + T_{\mathrm{s}}) - u_\beta(t) \cos(\omega T_{\mathrm{s}})}{\sin(\omega T_{\mathrm{s}})} \\ &= \frac{u(t + T_{\mathrm{s}}) - u(t) \cos(\omega T_{\mathrm{s}})}{\sin(\omega T_{\mathrm{s}})} \end{aligned} \tag{9}$$

*3.2. Figures, Tables and Schemes*

According to Equation (9), the value of the VQVS at the $k$th sampling can be obtained as.

$$u_\alpha(k) = \frac{u(k + 1) - u(k) \cos(\omega T_{\mathrm{s}})}{\sin(\omega T_{\mathrm{s}})} \tag{10}$$

where $k$ is the sampling point, $k = 1, 2, 3, \ldots$ .

Therefore, the quadrature voltage value can be obtained according to the grid voltage value $u(k)$ and $u(k + 1)$ delayed by one sampling period $T_s$. When the sampling period $T_s$ is set as 0.5 ms, 1 ms and 2.5 ms, respectively, the VQVS constructed by the STD are shown by the blue curves in the Figure 3. As can be seen from the figure, the VQVS constructed by the STD can follow the ideal quadrature signal very well even if the sampling period changes. As a result, the accuracy of the STD algorithm is higher than other algorithms.

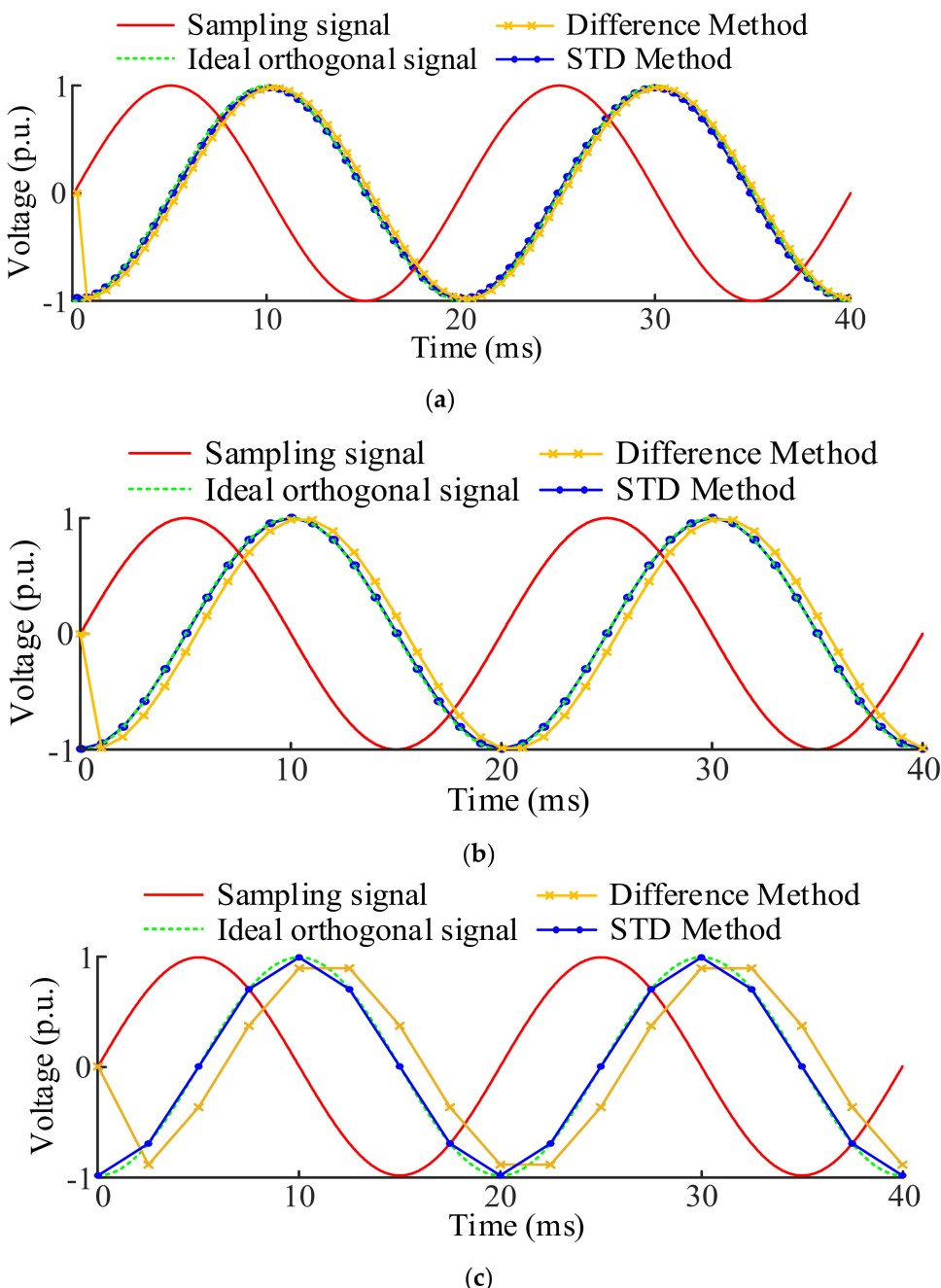

**Figure 3.** Effect of quadrature signal construction algorithm with different sampling periods (**a**) $T_s = 0.5$ ms; (**b**) $T_s = 1$ ms; (**c**) $T_s = 2.5$ ms.

### 3.3. Noise Sensitivity Analysis of the VQVS

In practical application, the voltage signal collected usually contains high frequency random noise, so it is necessary to investigate the effect of the VQVS on the high frequency random noise.

Assuming that the delay time is $\Delta T = n\,T_s$ ($n$ is the delay factor, its value is a non-negative integer), then the Equation (10) can be written as Equation (11) when the high frequency random noise is included.

$$
\begin{aligned}
\widetilde{u}_\alpha(k) &= \frac{\widetilde{u}(k+1) - \widetilde{u}(k)\cos(\omega\Delta T)}{\sin(\omega\Delta T)} \\
&= \frac{[u(k+1)+u_n(k+1)] - [u(k)+u_n(k)]\cos(\omega\Delta T)}{\sin(\omega\Delta T)} \\
&= u_\alpha(k) + u_{n\_\mathrm{STD}}(k)
\end{aligned}
\tag{11}
$$

where, the physical quantity with the superscript containing "~" indicates the inclusion of high-frequency random noise; $u_{n\_\mathrm{STD}}(k)$ is the VQVS of high frequency random noise.

$$
\begin{aligned}
u_{n\_\mathrm{STD}}(k) &= \frac{u_n(k+1) - u_n(k)\cos(\omega\Delta T)}{\sin(\omega\Delta T)} \\
&\leq \frac{1+\cos(\omega\Delta T)}{\sin(\omega\Delta T)}|u_n| = A_{\mathrm{STD}}|u_n|
\end{aligned}
\tag{12}
$$

where, ASTD is the amplification factor of the VQVS for noise under the worst conditions.

$$
A_{\mathrm{STD}} = \frac{1+\cos(\omega\Delta T)}{\sin(\omega\Delta T)} = \frac{1+\cos(\omega\cdot nT_s)}{\sin(\omega\cdot nT_s)}
\tag{13}
$$

Figure 4 shows the relationship between the ASTD and the sampling period $T_s$ for the difference and STD algorithms when $n = 1$. As can be seen from the figure, the ASTD from the STD algorithm is slightly lower than that of the difference algorithm when the $T_s$ period is the same.

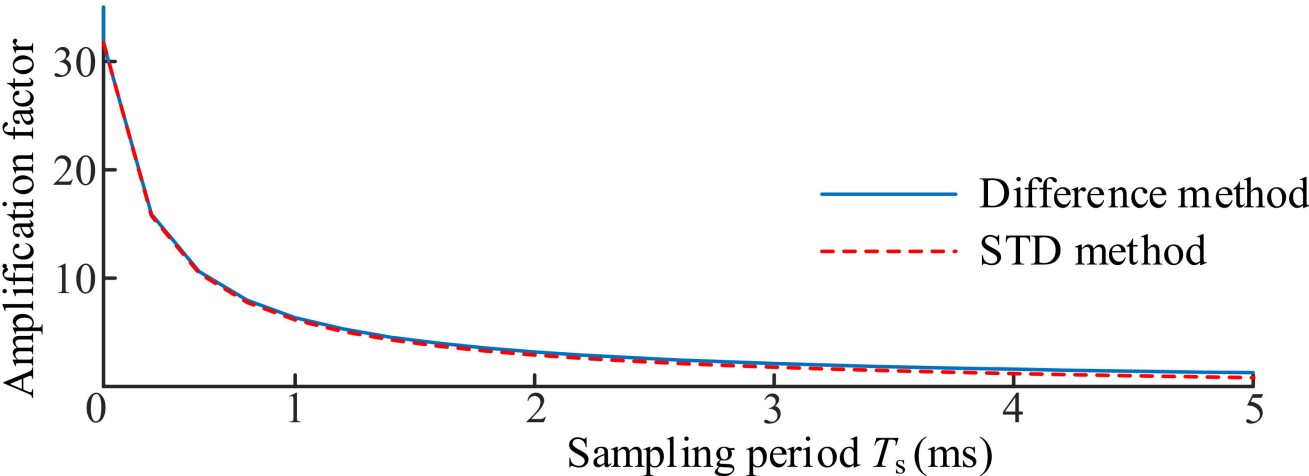

**Figure 4.** The relationship between Noise amplification factor and $T_s$ with different algorithms.

The parameters which related with the VQVS are ASTD, $\Delta T$, $n$ and $T_s$, and the relationship between them are shown in Figure 5. In the figure, the same color lines represented the same delay coefficients, the curves represent the relationship between ASTD and $T_s$, and the straight lines represent the relationship between $\Delta T$ and $T_s$ for different delay coefficients. As shown in Figure 5, ASTD decreases and $\Delta T$ increases when $n$ and $T_s$ increases. However, in order to improve detection speed and reduce noise, the values of the $n$ and $T_s$ can't be too large.

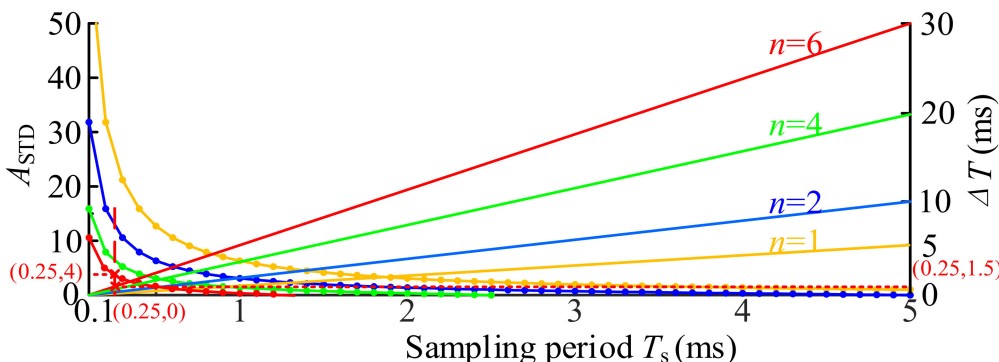

**Figure 5.** The relationship between ASTD and $T_s$ under different.

Comprehensive considering the relationship of these parameters, we set the $T_s = 0.25$ ms and $n = 6$ to investigate the characteristics of the VQVA. Figure 6 shows the VQVS constructed by different algorithms, the results shows that for the delay T/4 algorithm, there is obvious delay at the beginning of the curve from 0 to 5 ms even if the noise doesn't be amplified; for the difference algorithm, the noise is amplified; for the STD, the noise is slightly amplified, but there is no delay. When the sampling signal is filtered, the results of different algorithms are shown in the Figure 7. It can be seen that the STD has higher accuracy. Simultaneously, there is no delay. The above simulation results show that the STD has higher accuracy compared with the difference algorithm and delay T/4 algorithm. So the STD algorithm proposed in this paper shows a better performance in the response speed and accuracy.

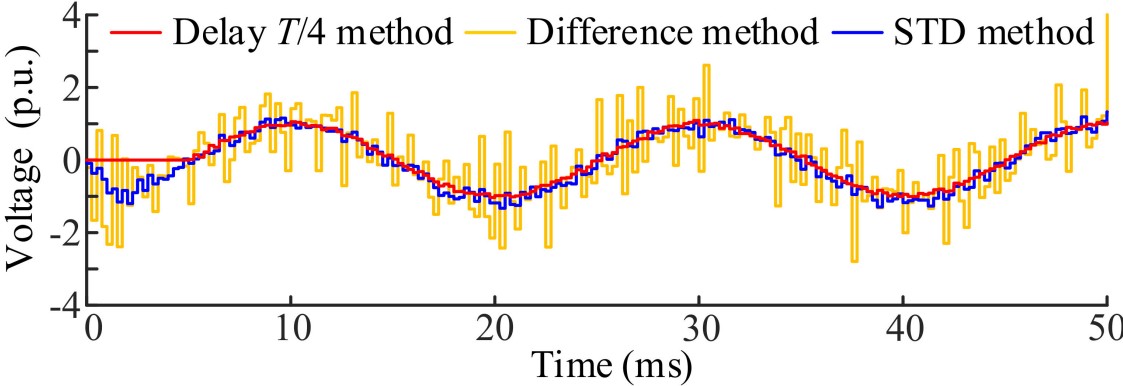

**Figure 6.** Noise sensitivity simulation of different quadrature signal construction algorithms without filter.

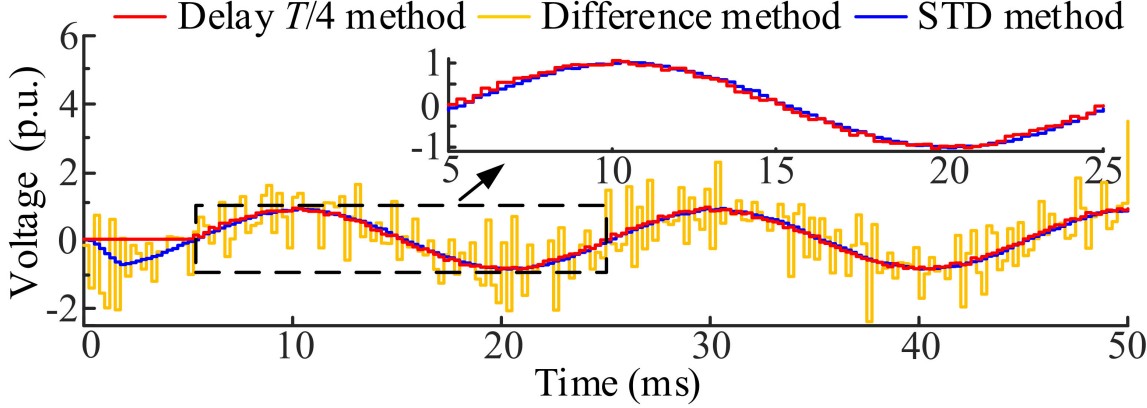

**Figure 7.** Noise sensitivity simulation of different quadrature signal construction algorithms with filter.

## 4. The Extraction of the Positive Sequence and Negative Sequence Voltage Components

Figure 8 shows the extraction step of the positive- and negative sequence voltage components for the three-phase unbalanced voltage sag. The three-phase grid voltage signal collected is decomposed into three independent single-phase voltage through a multiplexer, and then the VQVS which are separately orthogonal to the three single phases voltage signal are constructed by the STD. The three VQVSs and the voltage signals collected are divided into three groups to extract the positive- and negative sequence, respectively.

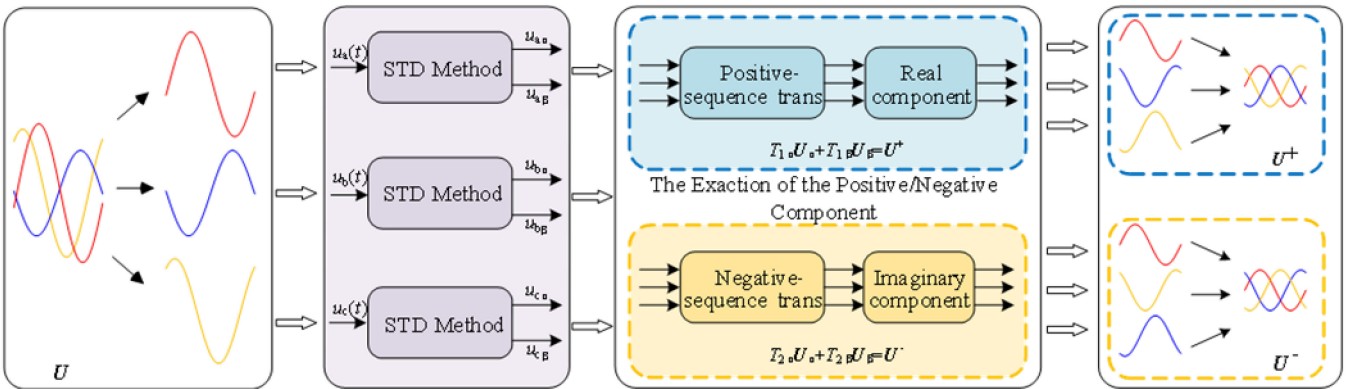

**Figure 8.** Schematic diagram of extracting positive-sequence and negative-sequence component.

In the two phase stationary coordinate system, the three-phase unbalanced voltage can be expressed by.

$$\dot{U} = U_\alpha + jU_\beta \tag{14}$$

$$\begin{cases} U_\alpha = \begin{bmatrix} u_{a\alpha}(t) & u_{b\alpha}(t) & u_{c\alpha}(t) \end{bmatrix}^T \\ U_\beta = \begin{bmatrix} u_{a\beta}(t) & u_{b\beta}(t) & u_{c\beta}(t) \end{bmatrix}^T \end{cases} \tag{15}$$

where, $\dot{U}$ is the three phase unbalanced grid voltage phasor; $U_\alpha$ and $U_\beta$ are the component of the voltage phasor in the $\alpha$-axis and $\beta$-axis; $u_{a\alpha}(t)$, $u_{b\alpha}(t)$ and $u_{c\alpha}(t)$ are the components of each phase voltage in the $\alpha$-axis; $u_{a\beta}(t)$, $u_{b\beta}(t)$ and $u_{c\beta}(t)$ are the components of each phase voltage in the $\beta$-axis.

For the orthogonal components $U_\alpha$ and $U_\beta$ in the Equation (14), it will be constructed by the STD algorithm. Here, the three-phase grid voltage signal is chosen as $U_\beta$, and another orthogonal $U_\alpha$ will be constructed by the STD.

The positive sequence component based on the symmetric component algorithm in the two-phase stationary coordinate system can be expressed by.

$$\dot{U}^+ = T_1 \dot{U} = U_\alpha{}^+ + jU_\beta{}^+ \tag{16}$$

$$\begin{cases} U_\alpha^+ = \begin{bmatrix} u_{a\alpha}^+(t) & u_{b\alpha}^+(t) & u_{c\alpha}^+(t) \end{bmatrix}^T \\ U_\beta^+ = \begin{bmatrix} u_{a\beta}^+(t) & u_{b\beta}^+(t) & u_{c\beta}^+(t) \end{bmatrix}^T \end{cases} \tag{17}$$

where, $\dot{U}^+$ is the positive sequence voltage component; $T_1$ is the positive sequence coordinate transformation matrix, and it can be expressed by.

$$T_1 = \frac{1}{3} \begin{bmatrix} 1 & a & a^2 \\ a^2 & 1 & a \\ a & a^2 & 1 \end{bmatrix} \tag{18}$$

According to Euler formula, the rotation factor $a$ and $a^2$ can be expressed by.

$$a = e^{j\frac{2\pi}{3}} = \cos\left(\frac{2\pi}{3}\right) + j\sin\left(\frac{2\pi}{3}\right) = -\frac{1}{2} + j\frac{\sqrt{3}}{2} \tag{19}$$

$$a^2 = e^{-j\frac{2\pi}{3}} = \cos\left(\frac{2\pi}{3}\right) - j\sin\left(\frac{2\pi}{3}\right) = -\frac{1}{2} - j\frac{\sqrt{3}}{2} \tag{20}$$

Combining the Equations (16), (17), (19) and (20), and the real component can be extracted, so the positive sequence component of the three-phase unbalanced grid voltage $\dot{u}^+$ can be derived as

$$\dot{u}^+ = T_{1\alpha}U_\alpha + T_{1\beta}U_\beta \tag{21}$$

where, $T_{1\alpha}$ and $T_{1\beta}$ are the coordinate transformation matrices of the positive sequence $\alpha$-component and $\beta$-component, respectively, and

$$T_{1\alpha} = \frac{1}{6}\begin{bmatrix} 2 & -1 & -1 \\ -1 & 2 & -1 \\ -1 & -1 & 2 \end{bmatrix}, \quad T_{1\beta} = \frac{\sqrt{3}}{6}\begin{bmatrix} 0 & -1 & 1 \\ 1 & 0 & -1 \\ -1 & 1 & 0 \end{bmatrix} \tag{22}$$

Similarly, the negative sequence component based on the symmetric component algorithm in the two phase stationary coordinate system can be expressed as

$$\dot{u}^- = T_2\dot{u} = U_\alpha^- + jU_\beta^- \tag{23}$$

$$\begin{cases} U_\alpha^- = \begin{bmatrix} u_{a\alpha}^-(t) & u_{b\alpha}^-(t) & u_{c\alpha}^-(t) \end{bmatrix}^T \\ U_\beta^- = \begin{bmatrix} u_{a\beta}^-(t) & u_{b\beta}^-(t) & u_{c\beta}^-(t) \end{bmatrix}^T \end{cases} \tag{24}$$

where, $\dot{u}^-$ is the negative-sequence voltage component; $T_2$ is the negative sequence coordinate transformation matrix, and it can be expressed as

$$T_2 = \frac{1}{3}\begin{bmatrix} 1 & a & a^2 \\ a & a^2 & 1 \\ a^2 & 1 & a \end{bmatrix} \tag{25}$$

Combining the Equations (19), (20), (23) and (24), and the real component can be extracted, so the negative sequence component of the three-phase unbalanced grid voltage $\dot{u}^-$ can be derived as

$$\dot{u}^- = T_{2\alpha}U_\alpha + T_{2\beta}U_\beta \tag{26}$$

where, $T_{2\alpha}$ and $T_{2\beta}$ are the coordinate transformation matrices of the negative sequence $\alpha$-component and $\beta$-component, respectively, and

$$T_{2\alpha} = \frac{1}{6}\begin{bmatrix} 2 & -1 & -1 \\ -1 & 2 & -1 \\ -1 & -1 & 2 \end{bmatrix}, \quad T_{2\beta} = \frac{\sqrt{3}}{6}\begin{bmatrix} 0 & 1 & -1 \\ -1 & 0 & 1 \\ 1 & -1 & 0 \end{bmatrix} \tag{27}$$

The positive sequence component $\dot{u}^+$ and negative sequence component $\dot{u}^-$ of the three-phase unbalanced voltage can be extracted by the PNST algorithm, the rotational components can be obtained by the synchronous coordinate transformation, then the dc component ($U_{d0}^+$, $U_{q0}^+$ and $U_{d0}^-$, $U_{q0}^-$) of the $dq$ coordinate system can be obtained. Combined with Equations (4) and (5), the amplitude and phase of the voltage can be obtained.

## 5. Simulations

To demonstrate the validity of the proposed algorithm, a MATLAB/Simulink model was established for the detection of the three-phase unbalance voltage sag. The simulation model are shown in Figure 9.

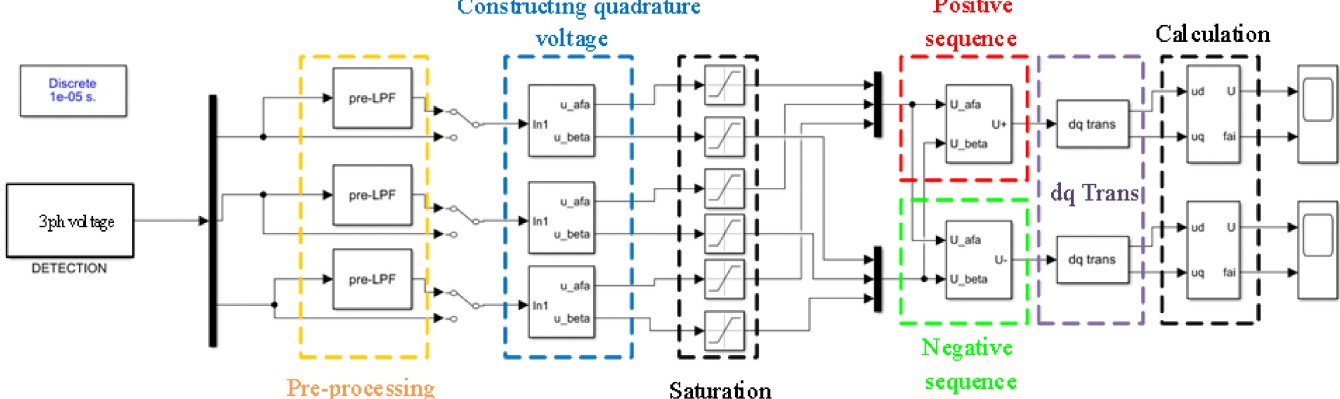

**Figure 9.** MATLAB/Simulink simulation model.

### 5.1. Transient-Response Analysis

Set the sampling frequency as 100 kHz, then the number of sampling points per cycle are 2000, the three-phase unbalanced voltage sag waveform can be obtained as the Figure 10. It can be seen that the three-phase voltages have different sag between 50 ms and 150 ms, the amplitude of A and B phases drop from 1 p.u. to 0.6 p.u., and the amplitude of C phase drops from 1 p.u. to 0.45 p.u. Besides, the phase of three phases has different jumps. Another thing needed to be noticed is that there are a lot of noise signal and harmonic in three-phase voltages, which will be filtered before detecting.

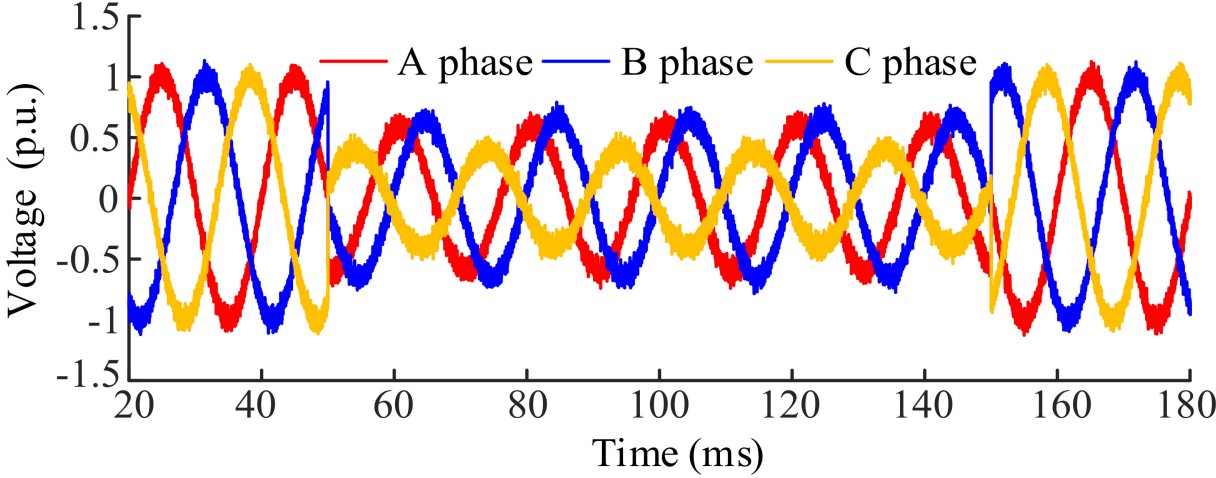

**Figure 10.** Three-phase unbalanced voltage sag drop waveform.

For the voltage sag, the positive- and the negative- sequence voltage component extracted by the proposed algorithm are shown in Figure 11. Compared with the waveform of the Figure 10, the amplitude of the positive- and negative sequence voltage components is stable and the frequency keep constant except for a short dynamic response time at the beginning and the end of the voltage sag. As a result, the proposed algorithm is effective. Simultaneously, it can also suppress the noise well.

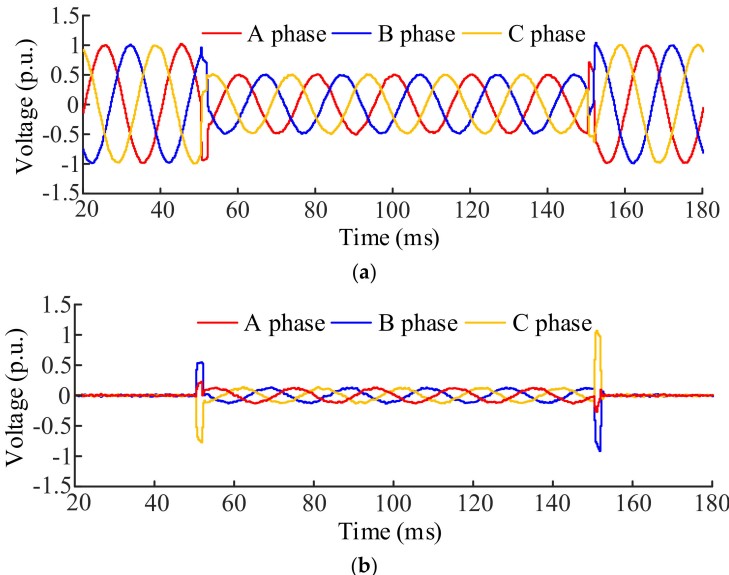

**Figure 11.** The extraction of positive-sequence and negative-sequence voltage component; (**a**) positive-sequence voltage component; (**b**) negative-sequence voltage component.

### 5.2. Sag-Detection Time

Comparing the proposed algorithm with the double *dq* transformation algorithm mentioned in [18]. the comparison results are showed in Figure 12. Here, Uideal, Ureference and Umanuscript are ideal voltage amplitude, double *dq* transformation amplitude and the article's voltage amplitude respectively. As shown in the Figure, the response speed of the proposed algorithm is faster than the double *dq* transformation algorithm mentioned in the literature no matter the positive sequence or negative sequence component. Simultaneously, it has little effect on the interference. So it can meet with the requirement of DVR better.

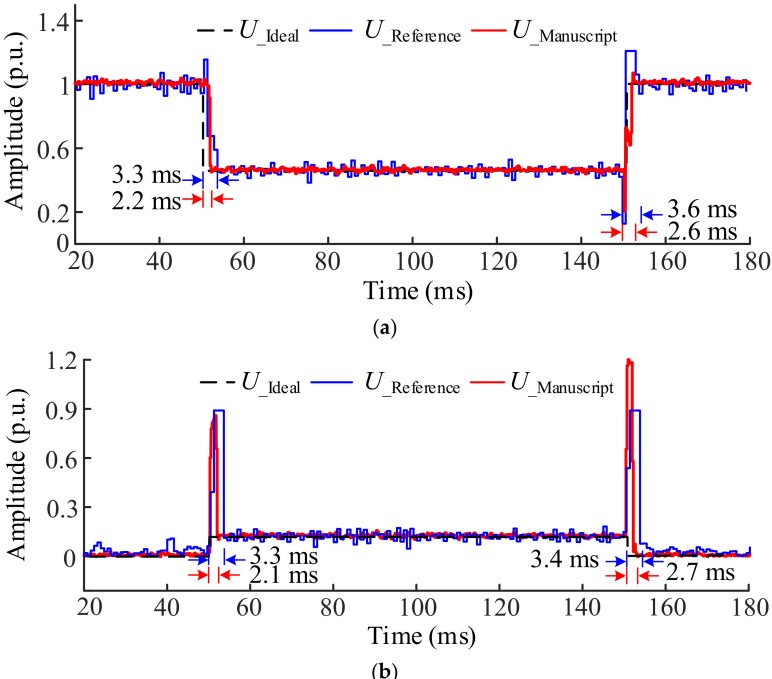

**Figure 12.** Comparison of positive-sequence and negative-sequence voltage amplitude detection results (**a**) positive-sequence component of three-phase voltage; (**b**) negative-sequence component of three-phase voltage.

## 6. Experiments

### 6.1. Hardware Setup

To validate the performance of the proposed detection algorithm for the three-phase unbalance voltage sag, a hardware test platform RT-LAB is setup in the laboratory [16]. Figure 13 shows the hardware configuration. The detection algorithm is performed by the DSP controller, the compilation and download of the simulation model are performed by the host machine, the target machine is connected to the Simulink real time system, the host machine is connected with the target machine through the slrtexplr tool, and the simulation results are output through the oscilloscope.

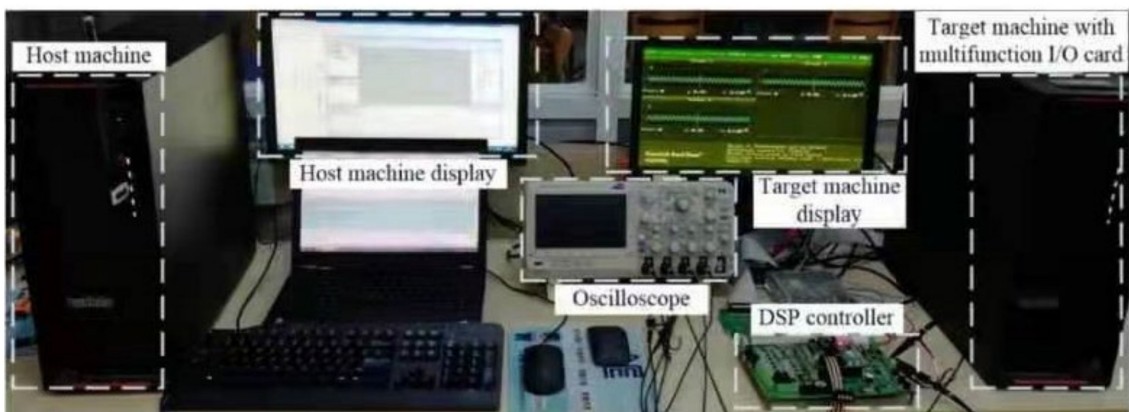

**Figure 13.** RT-LAB test platform.

### 6.2. Results and Analysis

The test parameters in the hardware setup are consistent with the model parameters in the Matlab/Simulink, and the voltage value is shown in the unit value. When a three-phase unbalanced voltage sag occurs, the sag waveform displayed in the oscilloscope is shown in the Figure 14. It can be seen that it is consistent with the Figure 10 of the simulation results. Besides, after filtering, three-phase voltage waveform becomes smoother, and voltage spikes is reduced a lot.

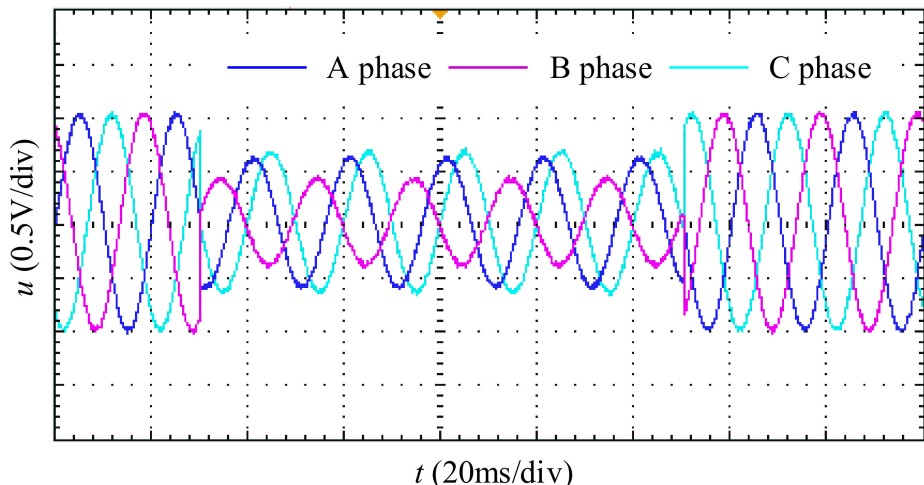

**Figure 14.** Three-phase unbalance voltage sag drop waveform in hardware-in-the-loop test.

For the positive- and negative sequence component of the three-phase unbalanced voltage sag, the results of the oscilloscope are shown in Figure 15. Compared with simulation result shown in Figure 11, due to inevitable environmental and hardware interference in the test, which cause some interference to the detection effect. According to the comparison

results, the waveform of the positive- and negative- sequence components shown in the oscilloscope is not as smooth as in the simulation, but it is still symmetric.

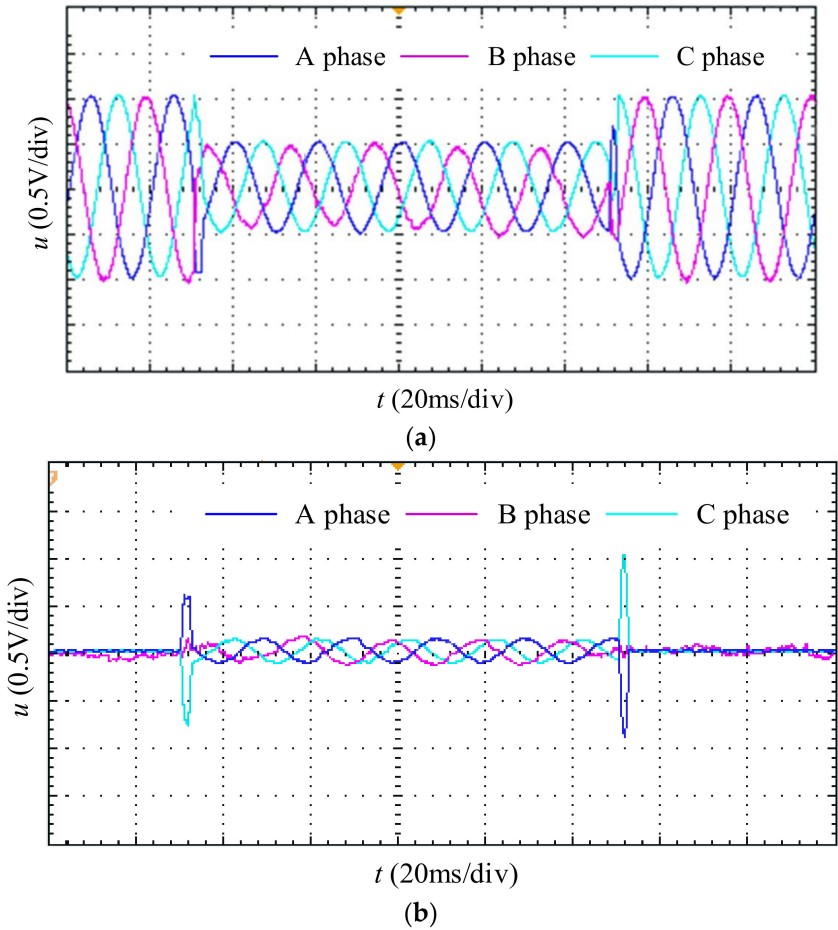

**Figure 15.** The results of extracting positive-sequence and negative-sequence voltage component in hardware-in-the-loop test; (**a**) positive-sequence voltage component; (**b**) negative-sequence voltage component.

For the positive- and negative-sequence component of the three-phase unbalanced voltage sag, the amplitude detection results of the oscilloscope are shown in Figure 16. Considering the signal delay of the processor and logic devices, the sag detection time in the test has 0.1–0.2 ms delay compared with that in the simulation, but the overall detection time is still within 3 ms for the proposed detection algorithm, which verifies the feasibility of the detection algorithm. In addition, during the dynamic response of the sag, the voltage amplitude variation detected by the detection algorithm is more stable, which shows better anti-interference ability compared with the reference algorithm.

In this paper, a new three-phase unbalanced voltage sag detection algorithm based on the STD and PNST is proposed to estimate voltage sag detection time. This algorithm adds a sampling cycle Ts to the $u_\alpha(t)$ and the $u_\beta(t)$, and setting the $u_\beta(t)$ as the sampling signal, then constructing the VQVS by changing the expression of the $u_\alpha(t + T_s)$ and the $u_\beta(t + T_s)$, which can accurately and effectively estimate the detection time of voltage sag. The validity and accuracy of the algorithm are confirmed by the Matlab/Simulink. The simulation results demonstrate that the influences of the white noise on voltage sag estimation can be effectively suppressed by the presented algorithm. Moreover, compared with the other reported detection algorithms, the proposed algorithm has the advantages of high detection accuracy, low operation cost, and strong robustness. A hardware platform is constructed to verify the feasibility and effectiveness of the proposed algorithm on the

embedded platform. Not only that, the voltage sag detection algorithm based on STD can provide the reference for electric utilities and users to evaluate power quality.

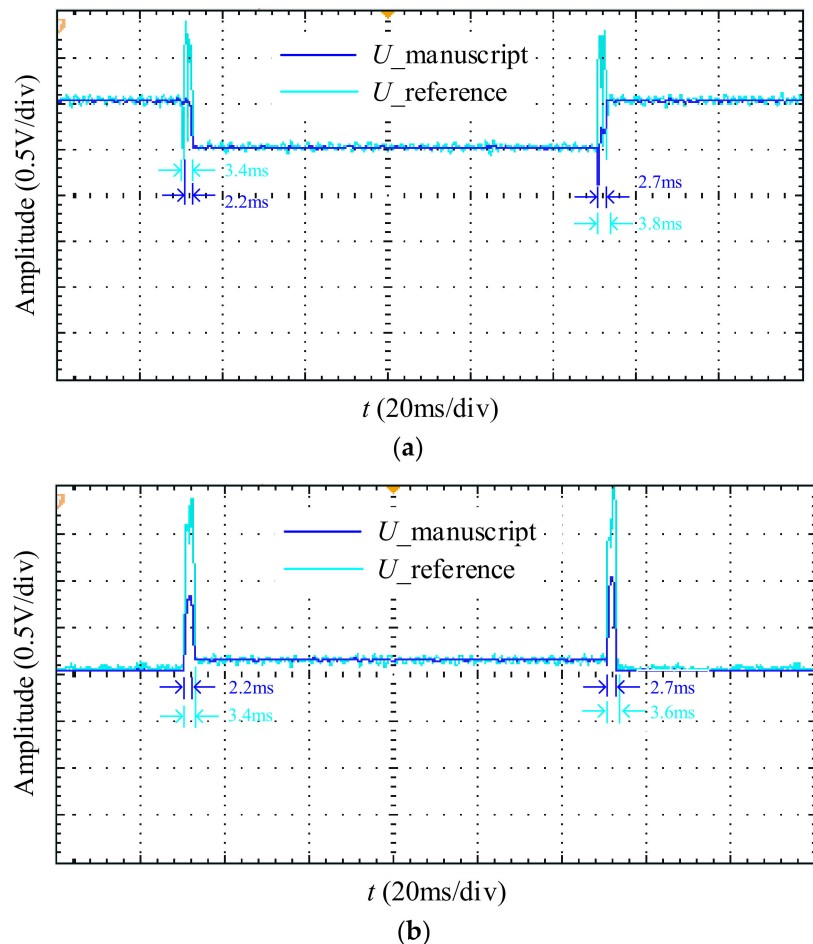

**Figure 16.** Comparison of positive-sequence and negative-sequence voltage amplitude detection results in hardware-in-the-loop test; (**a**) positive-sequence component; (**b**) negative-sequence component.

**Author Contributions:** Conceptualization, X.H.; Investigation, S.S.; Methodology, X.H.; Validation, S.J. and Y.W.; Writing—original draft, Y.L. All authors have read and agreed to the published version of the manuscript.

**Funding:** This research was funded by the Fundamental Research Funds for the Central Universities (No. ZYGX2019J039).

**Institutional Review Board Statement:** Not applicable.

**Informed Consent Statement:** Not applicable.

**Data Availability Statement:** The study did not report any data.

**Conflicts of Interest:** The authors declare no conflict of interest.

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
