# Peer review of "The Detection Algorithm Based on the Short Time Delay for Three-Phase Unbalanced Voltage Sag"

_electronics, doi:10.3390/electronics11172646_

Round 1

Reviewer 1 Report

Generally, a fairly paper that unfold a meritorious research effort, but with several shortcomings as follow:

-         -  mainly the Abstract is adequate;

-        -   the Introduction chapter begins with an oversized “V” letter, difficult to understand the reasons why; although this paragraph is extensive and fulfills the expectations for a journal paper;

-          - unfortunately during the full paper the text body flows together with the chapters denomination and is difficult to catch from where starts a new chapter or paragraph – it is recommended adequate blank lines insertion for delimitation;

-         -  in a similar way figure captures flows together with the text body (see the above recommendation);

-          - unexpected blank spaces leaved randomly during some pages;

-          - some figure captures are undersized (as example: fig. 4, 5, 6) and are difficult to understand;

-          - several equations are not centered or adequate aligned;

-          - fig. 9 is too small, in its actual form is incomprehensible;

-          - in a similar way the experimental results shown in fig. 11 and 12 are difficult to understand, the captures are undersized;

-          - the fig 13 that shows the experimental setup is in a similar way too small, please insert more detailed explanations regarding the hardware architecture and software technologies used;

-         -  the References list looks ok.

The paper may be accepted for journal publication after careful revision.

Author Response

Dear reviewer,

       Please find the reply in the attachment.

       Thank you very much.

Reviewer 2 Report

The authors presented a method for detecting three-phase unbalanced voltage sag.

 The contributions are sufficient for the manuscript to be published in Electronics, however, some comments are offered for improving this article.

 The manuscript’s strengths.

 1) The objective of the study is clear and matches the scope of the journal.

2) The manuscript is well organized and easy to read.

3) The authors have compared the current study with the available literatura.

4) An interesting aspect of this manuscript is the real-time operation assessment.

5) The system has been implemented.

 The manuscript’s weaknesses.

1) More information on the implementation of the algorithms would be welcome.

Author Response

Dear reviewer:

         Thank you for your question!

         You can find the  reply of  the question in the attachment. 

Round 2

Reviewer 1 Report

The paper now is suitable for journal publication.

Reviewer 2 Report

The revision is satisfactory. After incorporating of comments, remarks and recommendations from other reviewers and the editor, this article has an even higher quality. I recommend this article for publication in its current form.